# IL-6 inhibition prevents costimulation blockade-resistant allograft rejection in T cell-depleted recipients by promoting intragraft immune regulation in mice

Moritz Muckenhuber ®[1], Konstantinos Mengrelis[1], Anna Marianne Weijler ®[1], Romy Steiner[1], Verena Kainz[1], Marlena Buresch[1], Heinz Regele[2], Sophia Derdak[3], Anna Kubetz[1] & Thomas Wekerle ®[1] ✉

The efficacy of costimulation blockade with CTLA4-Ig (belatacept) in transplantation is limited due to T cell-mediated rejection, which also persists after induction with anti-thymocyte globulin (ATG). Here, we investigate why ATG fails to prevent costimulation blockade-resistant rejection and how this barrier can be overcome. ATG did not prevent graft rejection in a murine heart transplant model of CTLA4-Ig therapy and induced a pro-inflammatory cytokine environment. While ATG improved the balance between regulatory T cells (Treg) and effector T cells in the spleen, it had no such effect within cardiac allografts. Neutralizing IL-6 alleviated graft inflammation, increased intragraft Treg frequencies, and enhanced intragraft IL-10 and Th2-cytokine expression. IL-6 blockade together with ATG allowed CTLA4-Ig therapy to achieve long-term, rejection-free heart allograft survival. This beneficial effect was abolished upon Treg depletion. Combining ATG with IL-6 blockade prevents costimulation blockade-resistant rejection, thereby eliminating a major impediment to clinical use of costimulation blockers in transplantation.

Costimulation blockade is an established therapeutic alternative to calcineurin inhibitors (CNI) in transplantation[1–3]. Cytotoxic T lymphocyte-associated antigen 4 (CTLA4)-Ig, a recombinant extracellular domain of CTLA4 fused to a mutated human IgG1 Fc fragment, outcompetes CD28 for their shared ligands, CD80 (B7.1) and CD86 (B7.2)[4], thereby depriving T cells of costimulatory signals required for their activation. In clinical kidney transplantation, belatacept (a 2nd generation CTLA4-Ig with a mutated CTLA4 portion[5])-based regimens are associated with a lower incidence of *de-novo* donor-specific antibodies (DSA), superior graft function, and improved long-term patient and graft survival, compared with standard-of-care CNI regimens[2]. Yet, clinical use of CTLA4-Ig/belatacept—the first primary immunosuppressant approved since the introduction of CNIs four decades ago—remains very limited in kidney

transplantation and is almost non-existent in nonrenal transplantation[6,7]. One reason for the poor uptake of costimulation blockade is the increased rate of acute T cell-mediated rejection (TCMR) episodes when used with basiliximab (anti-CD25 mAb), the only induction agent approved for use with belatacept by FDA/EMA[6,8,9]. Mechanistically, two distinct T cell subsets have been implicated in costimulation blockade-resistant rejection (CBRR): regulatory T cells (Treg) and memory T cells. First, several experimental studies demonstrated that CTLA4-Ig compromises Treg homeostasis and their suppressive function[10,11]. Second, in non-human primate models[12], but also clinical kidney transplantation[13], specific CD28-independent CD8 memory T cell populations were found to be associated with CBRR[14]. Consequently, compensating for the negative effects on Tregs while controlling CD8

[1]Div. of Transplantation, Dept. of General Surgery, Medical University of Vienna, Vienna, Austria. [2]Clinical Institute of Pathology, Medical University of Vienna, Vienna, Austria. [3]Core Facilities, Medical University of Vienna, Vienna, Austria. ✉e-mail: thomas.wekerle@meduniwien.ac.at

memory T cells is an attractive therapeutic goal[15]. However, despite substantial effort, no clinically applicable therapeutic strategy that adequately controls CBRR has been established.

Downsizing the recipient's T cell pool with anti-thymocyte globulin (ATG) is routinely used to prevent TCMR in CNI-treated patients with high immunological risk[16,17]. ATG reduces TCMR frequencies by >40% when used with CNIs (cyclosporine or tacrolimus)[18]. Intriguingly, no similar effect is observed when ATG is used with costimulation blockade even though ATG has become the induction therapy most frequently used with belatacept[6]. Promising data from early small cohort studies[19] could not be confirmed in larger trials as TCMR frequencies under belatacept remained significantly elevated (≈25%) despite ATG induction, compared with CNI (≈7%)[20,21]. The mechanisms responsible for the observed lack of therapeutic efficacy have not been determined.

ATG is the purified polyclonal IgG fraction of rabbits sensitized with human thymocytes or T cells. It depletes >95% of peripheral blood T cells via antibody-dependent cell-mediated cytotoxicity, activation-induced cell death, and complement-dependent lysis. Importantly, Tregs are relatively spared from ATG-mediated depletion[22]. Therefore, ATG shifts the balance between Tregs and effector T cells in favor of Tregs. CTLA4-Ig itself decreases Treg frequencies, which likely contributes to increased rejection frequencies[23,24]. Therefore, combining CTLA4-Ig with ATG is attractive from a mechanistic point of view, as recent experimental data highlight the potential of Tregs to prevent CBRR. Our group demonstrated that interleukin-2 (IL-2) complexes restore Treg frequencies under CTLA4-Ig and significantly prolong cardiac allograft survival in mice[25]. A similar observation was recently published with IL-2 complexes and CTLA4-Ig in a murine type 1 diabetes model[26]. These reports provide proof-of-concept that a functionally restored Treg compartment is able to control costimulation blockade-resistant (allo)-immune responses.

In this study, we sought to delineate the immunological mechanisms by which costimulation blockade-resistant immune responses evade control by ATG in a mouse model of cardiac transplantation with CTLA4-Ig.

## Results

### ATG fails to improve the intragraft Treg: Teff balance
In naïve mice, ATG (0.15 mg on days 0 and 5) depleted >95% of peripheral blood CD4 and CD8 T cells. Tregs but also CD8 effector memory T cells (TEM; CD44$^{high}$ CD62L$^{low}$) were relatively spared from depletion (Suppl. Fig. 1a–c), in line with previous literature[27,28]. Peripheral blood CD4 and CD8 T cells remained lowered for 4 and 7 weeks, respectively (suppl. Fig. 1d). We then combined ATG with a previously described CTLA4-Ig maintenance treatment that was modeled after the clinically approved dosing regimen[24,25,29], employing a fully mismatched, relatively costimulation blockade-resistant donor-recipient strain combination (BALB/c to B6)[30,31] (Fig. 1a). ATG extended the survival of cardiac allografts under CTLA4-Ig (MST >100 days with vs 36 days without ATG, $p = 0.037$). Nonetheless, almost half of the grafts were rejected within the 100-day follow-up, mirroring the failure of ATG to prevent CBRR observed clinically[20,21] (Fig. 1b). Comparison between the spleen and the intragraft T cell compartment 14 days after transplantation (a time when all hearts were still beating) revealed that splenic Treg frequencies doubled upon ATG induction (Fig. 1c), shifting the Treg:CD8 T cell ratio in favor of Tregs (Fig. 1d). Strikingly, within cardiac allografts, ATG had no effect on Treg frequencies or Treg:CD8 T cell ratios (Fig. 1c, d). Thus, ATG reshapes the peripheral T cell compartment in favor of Tregs, but fails to do so within the allograft.

### ATG releases proinflammatory cytokines, including IL-6 whose perioperative blockade prevents costimulation blockade-resistant rejection
A potential mechanism for the lack of ATG efficacy is the release of pro-inflammatory cytokines. Indeed, ATG led to an increase in the pro-inflammatory serum cytokines interleukin-6 (IL-6), interferon gamma (IFNγ), interleukin-23 (IL-23) and interferon beta (IFNβ) 7 days after administration. This increase remained evident when CTLA4-Ig was given in addition to ATG (Fig. 1e, f). Thus, ATG leads to the release of pro-inflammatory cytokines that might counteract some of its beneficial effects of T cell depletion.

IL-6 has a prominent role in driving the alloresponse and is a key cytokine regulating the Treg: Teff balance[32,33]. Moreover, anti-IL6(R) mAbs are already in clinical use. We therefore focused on the increased levels of IL-6 following ATG administration and neutralized IL-6 (using blocking monoclonal antibodies; αIL6) in cardiac allograft recipients treated with ATG and CTLA4-Ig (Fig. 2a). Remarkably, adding IL-6 blockade prevented CBRR and led to 100% (8/8) long-term cardiac allograft survival under CTLA4-Ig plus ATG (Fig. 2b). Histopathologic graft analysis at day 14 after transplantation showed grade 2 or 3 rejections in 6/12 (50%) ATG/CTLA4-Ig-treated recipients, but in only 1/11 (9%) recipients treated with additional IL-6 blockade (ATG/CTLA4-Ig + αIL6) (Fig. 2c, d). IL-6 blockade on its own had no effect on cardiac allograft survival and did not further improve graft survival when added in CTLA4-Ig treated recipients. When combined with ATG (ATG + αIL6) graft survival was prolonged (MST = 31 days) compared to either substance alone (αIL6: MST = 11 days; ATG: MST = 16 days), but eventually most grafts were rejected within 45 days. Of note, adding IL-6 blockade to ATG/CTLA4-Ig decreased the serum levels of most inflammatory mediators quantified 7 days after transplantation, including those that were induced by ATG (IL-6, IL-23, IFNγ, and IFNβ) (suppl. Fig. 2a, b).

Serum IgG DSA were detectable in all recipients treated with ATG/CTLA4-Ig (8/8, 100%), but only in 3/8 (38%) mice treated with additional IL-6 blockade (Fig. 2e). Similarly, the MFI of recipient IgG binding to donor cells in vitro was significantly lower in recipients treated with additional αIL6 (Fig. 2f). Several reports suggest a crucial role for IL-6 during T follicular helper cell (Tfh) induction[34]. Interestingly, we did not observe an effect of αIL6 on the frequency (within CD4 T cells) or absolute count of Tfh (CD4$^+$ CD44$^+$ CXCR5$^+$ PD1$^+$) (suppl. Fig. 2c) or T follicular regulatory cells (Tfr; CD4$^+$ CD44$^+$ CXCR5$^+$ PD1$^+$ FOXP3$^+$) (suppl. Fig. 2d). Rather, perioperative IL-6 blockade interfered with DSA class-switch as MHC class I [H-2D$^d$] and class II [I-A$^d$] antigen-specific IgM was detectable in almost all recipients (ATG/CTLA4-Ig: 15/16; ATG/CTLA4-Ig + αIL6: 13/16) (suppl. Fig. 2e), whereas seroconversion from IgM to IgG was markedly reduced in αIL6 treated recipients (suppl. Fig. 2f, g).

Taken together, blocking IL-6, which is released upon ATG administration, prevents costimulation blockade-resistant rejection in cardiac allograft recipients treated with CTLA4-Ig and ATG.

### IL-6 blockade promotes intragraft regulation
Next, we investigated the specific effects of IL-6 blockade on the intragraft T cell compartment. We characterized graft-infiltrating leukocytes (GIL) via flow cytometry, immunofluorescent microscopy and RNA sequencing in cardiac allografts of recipients treated with or without αIL6 before the grafts were rejected (14 days after transplantation) (suppl. Fig. 3a). Overall, the number of infiltrating T cells was relatively low in both treatment groups (Fig. 3a). Within the intragraft T cell compartment, however, the fraction of CD8 TEM within CD8 T cells was significantly smaller in grafts isolated from αIL6 treated recipients (Fig. 3b). At the same time, intragraft Treg frequencies were significantly elevated through IL-6 blockade (Fig. 3c, left). Interestingly, Tregs specifically accumulated within cardiac allografts, as Treg levels in the spleen (Fig. 3c, right) and in peripheral blood (Fig. 5a) at this time point remained unaffected by the additional blockade of IL-6. Consequently, in grafts of αIL6 treated recipients, the ratio between Tregs and CD8 T cells was markedly shifted in favor of Tregs (Fig. 3d, left). Splenic Treg:CD8 T cell ratios—which were already

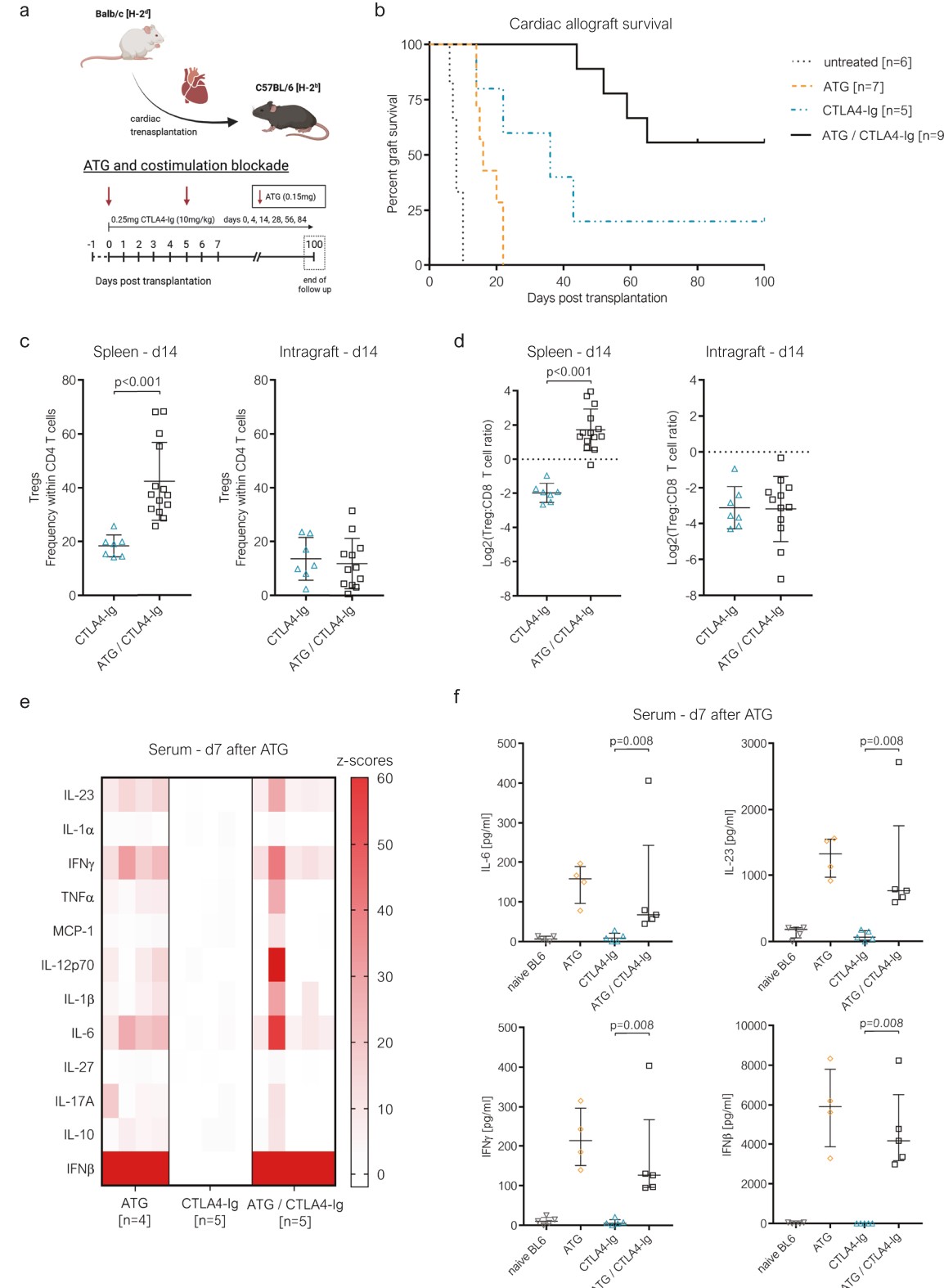

favoring Tregs with ATG alone—remained unaltered by additional IL-6 blockade (Fig. 3d, right).

We also visualized and quantified graft-infiltrating Tregs and CD8 T cells via immunofluorescent microscopy (Fig. 3e). Tregs but also CD8 T cells were frequently arranged in a perivascular pattern. The Treg:CD8 T cell ratio obtained via immunofluorescent microscopy was similar to the flow cytometry data and confirmed an accumulation of Tregs within grafts of recipients treated with IL-6 blockade (Fig. 3f).

Analysis of long-term surviving hearts (on day 100 after transplantation) revealed that at this late time point, the numbers of infiltrating CD4 and CD8 T cells were lower in mice treated with additional αIL6 (Fig. 3g). Intragraft CD8 TEM remained reduced in frequency and absolute number in αIL6-treated recipients (Fig. 3h). CD8 TEM formation was also impaired in the periphery in αIL6 treated recipients throughout the follow up (suppl. Figs. 3b, c). Notably, the Treg:CD8 T cell ratio remained increased in the graft in αIL6 treated recipients

**Fig. 1 | ATG re-shapes the composition of the peripheral, but not the intragraft T cell compartment. a–d** C57BL/6 (BL6) mice were grafted with a fully mismatched BALB/c cardiac allograft. Panel **a** illustrates the therapeutic regimen (*Note:* CTLA4-Ig is used as belatacept does not adequately bind murine CD80/CD86) (created with biorender.com). **b** Cardiac allograft survival under indicated immunosuppressive treatment regimens is depicted as Kaplan-Meier curves. Panel **c** illustrates the frequency of regulatory T cells (Tregs, CD4+ FOXP3+) within CD4 T cells in spleens (left) and cardiac allografts (right) isolated 14 days after transplantation (CTLA4-Ig [$n = 7$], ATG/CTLA4-Ig [$n = 14$]). Panel **d** shows the ratio between Tregs and CD8 T cells (log2 transformed) within spleens (left) and cardiac allografts (right) isolated 14 days after transplantation (CTLA4-Ig [$n = 7$], ATG/CTLA4-Ig [$n = 14$]). **e** and **f** C57BL/6 mice received two doses of anti-thymocyte globulin (ATG, 0.15 mg on days 0 and 5) with or without costimulation blockade or CTLA4-Ig alone (without cardiac transplantation). A panel of pro-inflammatory cytokines was quantified in sera obtained 7 days after the first ATG injection using a flow cytometry-based multiplex assay. **e** Cytokine levels are depicted descriptively as z-scores in a heat map. Each column represents an individual mouse. Panel **f** illustrates serum concentrations (pg/ml) of interleukin (IL)−6 (top, left), IL-23 (top, right), interferon-gamma (IFNγ, bottom, left) and interferon-beta (IFNβ, bottom, right), 7 days after the first ATG administration (naïve BL6 [$n = 4$], ATG [$n = 4$], CTLA4-Ig [$n = 5$], ATG/CTLA4-Ig [$n = 5$]). Each symbol represents an individual mouse. Lines indicate group means ± SD (**c** and **d**) or median ± IQR (**f**). Group comparisons (indicated with brackets) were conducted via two-sided unpaired t-test for normal distributed values (**c** and **d**) and two-sided Mann–Whitney-U test for variables not following normal distribution **f**.

also at this late time point (Fig. 3i), and indeed the difference compared to grafts without αIL6 even widened, demonstrating that the favorable intragraft Treg:Teff balance in αIL6 treated recipients persisted long-term.

To directly test whether the extension of cardiac allograft survival in recipients treated with additional IL-6 blockade (ATG/CTLA4-Ig + αIL6) was Treg-mediated, we depleted Tregs before (early Treg depletion) or 4 weeks after (late Treg depletion) transplantation using an anti-CD25 mAb (clone: PC61, 0.25 mg per dose)[24,25] (Fig. 4a). Depletion was confirmed via flow cytometry (suppl. Fig. 3d) and resulted in a >95% reduction in CD4+ CD25+ cells (and a > 75% reduction in peripheral blood CD4+ FOXP3+ cells). The majority of cardiac allografts were lost within 30 days after early Treg depletion at the time of transplant. Upon late Treg depletion 1 month post-transplant >50% of grafts were lost within the follow-up period (Fig. 4b). Hence, prolongation of cardiac allograft survival by IL-6 blockade is dependent on Tregs in our model.

To investigate whether neutralizing IL-6 influenced intragraft signaling patterns, we performed bulk RNA sequencing of flow-sorted GIL (7AAD- CD45+) isolated from cardiac allografts 14 days after transplantation. Gene set enrichment analysis (GSEA) revealed a significant enrichment of Treg-associated genes in graft infiltrating leukocytes of recipients treated with additional αIL6 compared to recipients treated with ATG/CTLA4-Ig (Fig. 4c), consistent with the increased intragraft Treg levels found by flow cytometry. Furthermore, gene sets representing interleukin-4/13 (IL-4/13) signaling and interleukin-10 (IL-10) signaling were significantly enriched upon IL-6 blockade (Fig. 4d). In line with these findings, using immunofluorescence microscopy (suppl. Fig. 4a), we observed a significantly higher number of IL-10 secreting CD4+ cells in cardiac allografts of recipients treated with additional IL-6 blockade compared to the ones under ATG/CTLA4-Ig (suppl. Fig. 4b).

IL-6 is known to drive Th17 lineage commitment of naïve T cells[35,36] and Th17 cells have been implicated in CBRR[37,38]. We therefore asked whether impairment of Th17-mediated immunity could explain the effects of αIL6 in our model. However, genes representing IL-17 signaling were not enriched in either of the two groups in the GSEA (nominal p = 0.334; suppl. Fig. 4c). Furthermore, the number of IL-17 positive CD4 + T cells within cardiac allografts, analyzed via immunofluorescent microscopy, was low in general and did not relevantly differ between the two treatment groups (suppl. Fig. 4d). Hence, inhibiting Th17 cell-mediated cardiac allograft injury seems not to be a critical mechanism by which IL-6 blockade prevents CBRR in this model.

Taken together, the effect of IL-6 blockade depends on Tregs, promotes Treg frequencies and Treg gene signatures within cardiac allografts and shifts signaling patterns of GIL towards Th2 cytokines, including IL-10.

## IL-6 blockade affects the Treg phenotype under ATG/CTLA4-Ig therapy

We then assessed the impact of IL-6 neutralization on the peripheral Treg compartment. IL-6 blockade increased peripheral blood Treg frequencies 8 days after transplantation in recipients treated with ATG and CTLA4-Ig (Fig. 5a). While Treg levels quickly declined thereafter, FOXP3 expression in Tregs remained upregulated for 3 weeks (Fig. 5b). 14 days after transplantation (Fig. 5c–k) expression of CD44 (Fig. 5c), Ki-67 (Fig. 5d) and CD25 (Fig. 5e) was significantly higher in splenic Tregs of transplant recipients treated with additional IL-6 blockade, while CTLA4 (Fig. 5f), PD-1 (Fig. 5g), and Helios (Fig. 5h) expression was similar to recipients that were not treated with αIL6. In this context, CD44 is known as a marker for effector Tregs with higher suppressive capacity[39], while CD25 expression was shown to correlate with Treg tissue stability[40].

Next, we performed UMAP-based dimensional reduction of these 6 functional Treg markers using a concatenated file including 90,000 Tregs (Fig. 5i). Automated clustering using FlowSOM[41] identified 5 distinct Treg phenotypes based on the expression profile of the 6 functional markers (Fig. 5j). As addition to quantifying the expression of each individual marker in global Tregs (as done in Fig. 5c–h), this allowed us to investigate whether specific subsets of phenotypically distinct Tregs were expanded or decreased in the periphery through IL-6 blockade. Thereby we found that a discrete subset of Tregs expressing the highest levels of PD-1 and CD44 (corresponding to an effector phenotype[42]) was specifically expanded upon IL-6 blockade (Fig. 5k). Taken together, IL-6 neutralization in cardiac allograft recipients treated with ATG and CTLA4-Ig has a stabilizing effect on peripheral Tregs and specifically induces a subset of Tregs highly expressing PD-1 and CD44. Thus, IL-6 neutralization impacts peripheral and intragraft Tregs.

## IL-6 blockade reverses intragraft inflammation observed under ATG/CTLA4-Ig

Finally, to investigate the effects of IL-6 neutralization on the graft itself, we performed bulk RNA sequencing of cardiac allografts explanted 14 days after transplantation, which revealed 48 protein-coding differentially expressed genes (DEG) between recipients treated with (ATG/CTLA4-Ig + αIL6, $n = 8$) or without (ATG/CTLA4-Ig, $n = 8$) additional IL-6 blockade (Fig. 6a, b). 35 DEG could be ontologically and functionally annotated based on their expression in cardiomyocytes and cardiac fibroblasts (13 DEG), endothelial cells (2 DEG) or (graft infiltrating) leukocytes (GIL; 20 DEG) (Table 1). Among cardiomyocyte and cardiac fibroblast-associated genes, only Elastin (*Eln*) was upregulated through IL-6 blockade. Interestingly, Elastin expression was shown to be repressed by inflammatory cytokines like tumor necrosis factor-alpha (TNFα) and interleukin-1 beta (IL-1β) and induced by transforming growth factor beta (TGFβ)[43]. Several other cardiomyocyte-associated genes linked with ischemia-reperfusion injury (IRI), inflammation and cardiac distress were downregulated through IL-6 blockade (Table 1)[43–51]. Also, endothelial *Cysltr2* encoding the cysteinyl leukotriene receptor 2 (CysLT2R), typically upregulated in endothelial cells upon IRI[44], was downregulated by IL-6 blockade. Leukotrienes are potent inflammatory mediators that, upon engagement with endothelial Cyslt2r promote trans-endothelial migration of

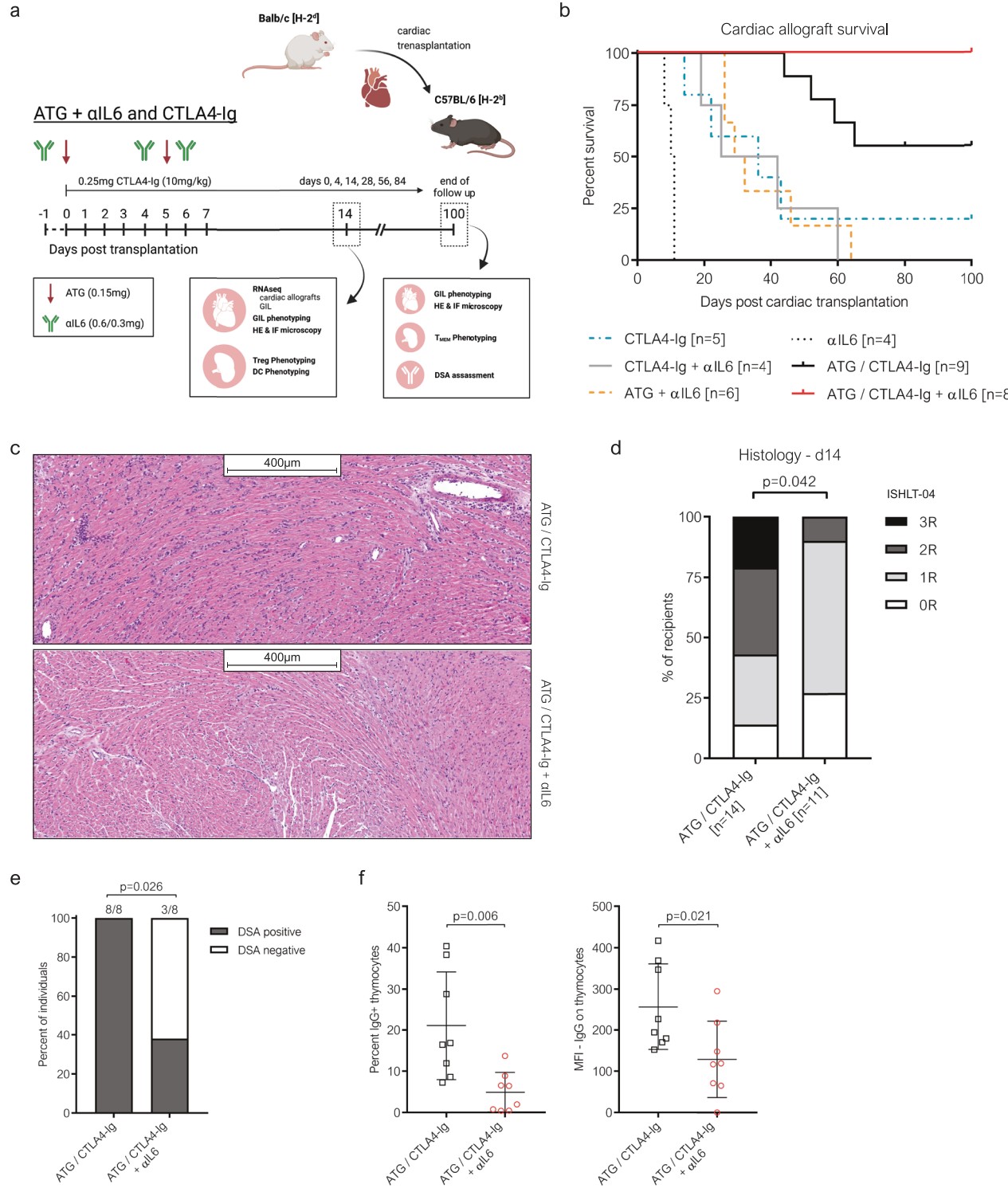

**b** Cardiac allograft survival

- CTLA4-Ig [n=5]
- αIL6 [n=4]
- CTLA4-Ig + αIL6 [n=4]
- ATG / CTLA4-Ig [n=9]
- ATG + αIL6 [n=6]
- ATG / CTLA4-Ig + αIL6 [n=8]

**d** Histology - d14

**e** p=0.026

**f** p=0.006    p=0.021

leukocytes[52]. In line with these findings, genes associated with (graft-infiltrating) leukocytes were all downregulated in cardiac allografts of αIL6 treated recipients, in accordance with fewer leukocytes being present in these grafts. Notably, cc-chemokine ligand 28 (CCL28), a potent chemokine for C-C chemokine receptor type 10 (CCR10) expressing Tregs, which is inducible in endothelial cells[53,54], was upregulated in cardiac allografts of αIL6 treated recipients. Gene set enrichment analyses demonstrated that gene sets representing an adaptive immune response (Fig. 6c, top) and T cell-mediated cyto-toxicity (Fig. 6c, bottom) were significantly downregulated in recipients treated with additional IL-6 blockade (ATG/CTLA4-Ig + αIL6)

compared with control animals (ATG/CTLA4-Ig). These data indicate that IL-6 blockade attenuates early inflammation within cardiac allografts.

## Discussion

Costimulation blockade as immunosuppression has several benefits over the current standard of CNIs and could improve long-term outcome after kidney transplantation. However, this potential has not been realized so far, in part due to the higher rejection rates observed with costimulation blockade. With this current study, we sought to delineate the mechanisms responsible for the failure of ATG induction,

**Fig. 2 | IL-6 blockade prevents costimulation blockade-resistant allograft rejection.** Panel **a** represents a schematic illustration of the experimental setup and the immunosuppressive treatment regimen combining anti-thymocyte globulin (ATG) and perioperative interleukin (IL)−6 blockade under CTLA-4-Ig (created with biorender.com). **b** C57BL/6 mice received a fully mismatched BALB/c cardiac allograft under the indicated immunosuppressive treatment regimen (*Note:* CTLA4-Ig and ATG/CTLA4-Ig groups are also shown in panel 1b). Kaplan-Meier curves depict the cardiac allograft survival for each treatment group. Cardiac allograft survival was compared between groups using a two-sided log-rank test. In selected recipients, cardiac allografts were explanted 14 days after transplantation for histological examination. Panel **c** shows two representative images of the hematoxylin-eosin (HE) stained cardiac allograft tissue sections generated for the histological analyses presented in panel **d. d** Rejection severity was graded according to the 2004 International Society for Heart and Lung Transplantation (ISHLT-04) cardiac allograft rejection scores depicted as contingency bar graph. ISHLT scores were compared between the two groups via two-sided Mann-Whitney U test. At rejection or at the end of the follow up (day 100), presence of serum IgG donor-specific antibodies (DSA) was assessed via flow crossmatch **e** and **f**. Panel **e** illustrates the fraction of recipients with detectable IgG DSA (DSA positive) within each treatment group. The rates of DSA positive recipients were compared between the two treatment groups using Fisher's exact test. Panel **f** depicts the frequency of IgG+ donor thymocytes (left) and the median fluorescence intensity (MFI) of IgG on donor thymocytes (right) for each individual recipient, analyzed using a two-sided unpaired t-test (ATG/CTLA4-Ig [$n = 8$], ATG/CTLA4-Ig + αIL6 [$n = 8$]). Each symbol represents an individual mouse. Lines indicate group means ± SD.

which is effective in reducing rejection rates when used with CNIs, to control costimulation blockade-resistant rejection[20,21]. Using a mouse heart transplant model of CTLA4-Ig maintenance therapy[24,25,29], we found that ATG fails to improve the Treg: Teff balance within the allograft, which is improved only in the peripheral compartment. Cytokine measurements pointed to elevated serum IL-6 levels as a potential cause for the lack of ATG efficacy. Indeed, neutralizing IL-6 attenuated intragraft and systemic inflammation, allowed Tregs to accumulate in cardiac allografts, and ultimately prevented CBRR. Depleting Tregs triggered graft loss, directly showing that the extension of graft survival mediated by IL-6 blockade is Treg-dependent. Elevated intragraft Treg frequencies occurred early after transplantation, persisted until the end of follow up, and were accompanied by a shift of signaling patterns in graft-infiltrating leukocytes towards IL-10 and other Th2 cytokines. Our results provide a mechanistic explanation for the lack of efficacy of ATG induction in controlling CBRR and reveal IL-6 blockade as therapeutic intervention overcoming this problem.

CBRR is typically caused by a "breakthrough" of CD28-independent CD8+ effector T cells[55]. Recent reports demonstrate that Tregs can control costimulation-blockade-resistant immune responses in autoimmunity and transplantation if the negative effects of CTLA4-Ig on the Treg compartment are compensated[25,26]. In this context, Tregs residing within the allograft are of particular interest as landmark studies by Waldmann and colleagues identified intragraft Tregs in tolerized skin allografts[56] and furthermore demonstrated that peripherally induced Tregs accumulate in tolerated grafts to actively prevent rejection[57,58]. More recent data arising from the development of chimeric antigen receptor (CAR) Tregs also indicate that donor-antigen specific Tregs, accumulating within allografts, are particularly effective in suppressing alloimmunity[59]. ATG's main mechanism of action is to deplete effector T cells, while partially preserving Tregs. Thereby, ATG shifts the Treg: Teff balance in favor of Tregs[22]. In theory, ATG should thereby reinforce Treg-mediated control over effector T cells. Our results, however, show that within cardiac allografts, such a favorable shift does not occur. Thus, ATG fails to reinstate Treg-mediated control over effector T cells within cardiac allografts under CTLA4-Ig.

Our results also put forward a likely explanation as to why Tregs do not accumulate within cardiac allografts following ATG: We found that ATG triggers the secretion of several pro-inflammatory cytokines. Thereby, together with other known triggers, such as ischemia-reperfusion injury and surgical trauma, ATG contributes to an inflammatory environment after transplantation. Among the cytokines induced by ATG (IL-6, IL-23, IFNγ, and IFNβ), IFNγ has multifaceted mechanisms: While pro-inflammatory properties of IFNγ, including the upregulation of class I and II MHC and enhanced T cell infiltration, promote allograft rejection[60], IFNγ has also been shown to be crucial for Treg-mediated suppression of allo-immune responses, particularly within the allograft[61,62]. Similarly, IFNβ was recently demonstrated to

stabilize Tregs via FOXP3 acetylation[63]. Blocking IL-23 (and interleukin-12) has demonstrated partial therapeutic efficacy when combined with CTLA4-Ig in a skin graft model[64]. As IL-6 has a prominent role in driving the alloresponse[32,65] and is a key cytokine regulating the Treg: Teff balance[33] we focused on IL-6 as a therapeutic target. We show that neutralizing IL-6 mitigates ongoing inflammatory responses within cardiac allografts and allows Tregs to accumulate. Interestingly, we observed that blocking IL-6 also decreased the serum levels of most other inflammatory mediators that we assessed following transplantation, including the ones that we found to be specifically induced by ATG (IL-6, IL-23, IFNγ, and IFNβ).

Negative effects of IL-6 on peripheral Treg induction are well established within the literature[66-68]. Consequently, several studies investigating IL-6 deficiency in autoimmunity[69], but also in transplantation[70-73] have been reported. Besides ischemia-reperfusion injury and alloimmunity, hyperlipidemia was recently identified as a driver of IL-6 release in transplantation, causing Treg dysfunction and accelerated allograft rejection[74]. In clinical kidney transplantation, a team of investigators led by Vincenti assessed the potential benefit of IL-6 receptor blockade (using tocilizumab) in stable recipients (on CNIs) with subclinical inflammation in protocol biopsies. Tocilizumab improved BANFF total inflammation scores in follow-up biopsies and increased circulating Treg levels[75].

The direct effects of IL-6 on stable lineage-committed Tregs remain more controversial. Recent data indicate that under certain conditions, IL-6 might in fact enhance Treg proliferation in a TNFR2-dependent manner[76,77]. In our model, IL-6 blockade resulted in only a modest increase in systemic Treg levels, which were already elevated after ATG induction. Nonetheless, we identified distinct phenotypical changes in Tregs of αIL-6 treated recipients that may correspond to a more suppressive Treg phenotype[78].

Moreover, we find that the mechanisms of ATG and αIL6 complement each other. It was shown that a functionally restored Treg compartment can control costimulation blockade-resistant immune responses[25,26]. ATG re-instates Treg-mediated control over effector T cells in the peripheral lymphoid compartment, but fails to do so within cardiac allografts − resulting in graft rejection. We show that additional IL-6 blockade mitigates intragraft inflammation and thereby allows Tregs, which are enriched within the recipient's downsized T cell pool following ATG, to accumulate within the allograft. IL-6 blockade thereby resolves the imbalance between peripheral and intragraft Treg levels that we observed following ATG induction alone and allows extending Treg-mediated control over effector T cells to the intragraft compartment. The ATG/CTLA4-Ig/αIL6 drug combination thereby controls alloimmunity through immunomodulatory mechanisms that heavily rely on regulation rather than the mere suppression of effector T cells. Ultimately, our data suggest that protocols, which rely more actively on regulation as a mechanism of action might allow for a better control over CBRR than currently approved CTLA4-Ig/belatacept drug combinations.

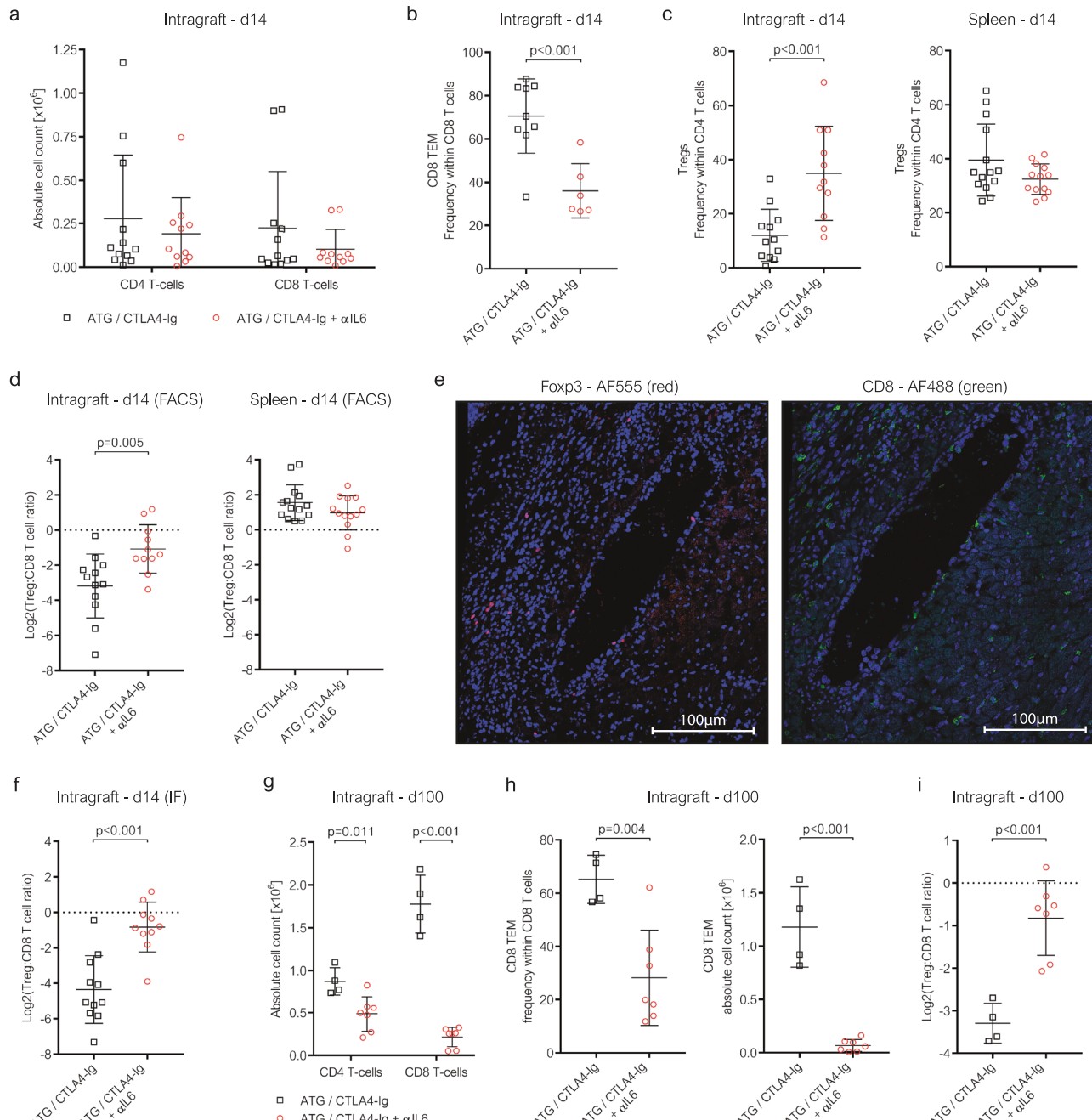

**Fig. 3 | IL6 blockade leads to intragraft accumulation of Tregs.** C57BL/6 mice were transplanted with a fully mismatched BALB/c cardiac allograft. 14 days after transplantation, cardiac allografts were explanted in recipients treated with indicated immunosuppressive regimens. **a–d** Grafts were enzymatically digested for flow cytometric analysis of graft infiltrating leukocytes (GIL). **a** The absolute number of graft infiltrating CD4 and CD8 T cells was calculated based in the total number of GIL isolated per cardiac allograft (ATG/CTLA4-Ig [n = 12], ATG/CTLA4-Ig + αIL6 [n = 11]). **b** The frequency of CD8 effector-memory T cells (CD8 TEM, CD8+ CD44high CD62Llow) within infiltrating CD8 T cells was quantified (ATG/CTLA4-Ig [n = 9], ATG/CTLA4-Ig + αIL6 [n = 6]). Panel **c** depicts the frequency of regulatory T cells (Tregs, CD4+ FOXP3+) within CD4 T cells in cardiac allografts (left) and spleens (right) (Intragraft: ATG/CTLA4-Ig [n = 12], ATG/CTLA4-Ig + αIL6 [n = 11]; Spleen: ATG/CTLA4-Ig [n = 14], ATG/CTLA4-Ig + αIL6 [n = 13]). Panel **d** shows the ratio between Tregs and CD8 T cells (determined via flow cytometry; log2 transformed) within cardiac allografts (left) and spleens (right) (Intragraft: ATG/CTLA4-Ig [n = 12], ATG/CTLA4-Ig + αIL6 [n = 11]; Spleen: ATG/CTLA4-Ig [n = 14], ATG/

CTLA4-Ig + αIL6 [n = 13]). Panel **e** shows a representative image of cardiac allograft tissue sections stained for FOXP3 (left) and CD8 (right) obtained via immunofluorescence microscopy for the quantification of Tregs and CD8 T cells presented in panel **f**. **f** The number of Tregs (FOXP3+, red) and CD8 T cells (CD8+, green) per mm² was quantified in these tissue sections of cardiac allografts. The ratio between infiltrating Tregs and CD8 T cells was calculated and is depicted in log2-transformed form (ATG/CTLA4-Ig [n = 11], ATG/CTLA4-Ig + αIL6 [n = 10]). **g–i** Long-term surviving cardiac allografts were explanted 100 days after transplantation and enzymatically digested (ATG/CTLA4-Ig [n = 4], ATG/CTLA4-Ig + αIL6 [n = 7]). Graft infiltrating leukocytes were analyzed via flow cytometry to quantify the infiltrating CD4 and CD8 T cells **g** and CD8 TEM **h**. **i** The ratio of infiltrating Tregs and CD8 T cells is depicted in log2-transformed form. Each symbol represents an individual mouse. Lines indicate group means ± SD. All group comparisons (indicated with brackets) were conducted via two-sided unpaired t-tests. Representative flow cytometry data are provided in (suppl. Fig. 5).

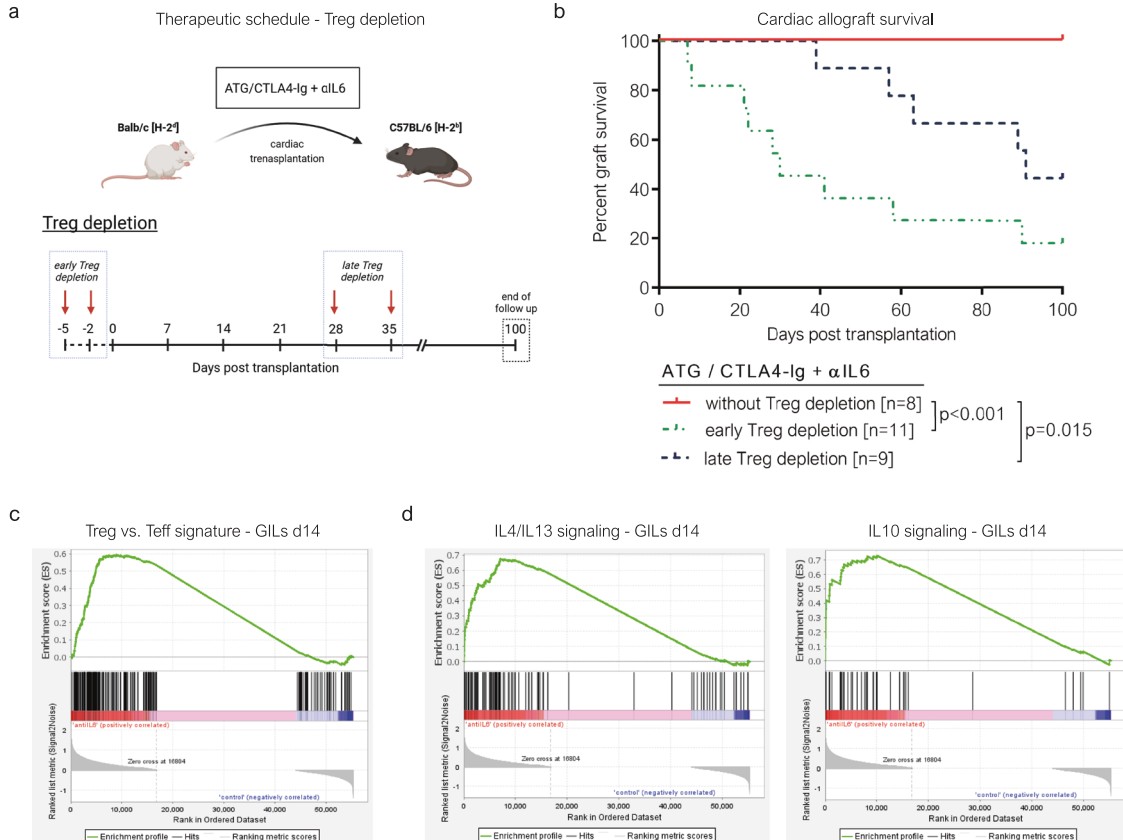

**Fig. 4 | IL6 blockade-mediated extension of cardiac allograft survival is Treg-dependent.** Panel **a** represents a schematic illustration of the experimental setup and the time schedule for regulatory T cell (Treg) depletion (created with bior-ender.com). **b** C57BL/6 mice received a fully mismatched BALB/c cardiac allograft under ATG/CTLA4-Ig + αIL6. In these recipients, Tregs were depleted using a monoclonal anti-CD25 antibody (clone: PC61; 0.25 mg per dose) 5 and 2 days before (early Treg depletion) or 28 and 35 days after (late Treg depletion) cardiac trans-plantation. Kaplan-Meier curves depict the cardiac allograft survival for each group (*Note:* "ATG/CTLA4-Ig + αIL6 without Treg depletion" group is also shown in panel 2b). Cardiac allograft survival was compared between groups using a two-sided log-rank test. **c, d** In selected recipients (ATG/CTLA4-Ig with or without αIL6, without Treg depletion), cardiac allografts were explanted 14 days after trans-plantation, enzymatically digested, and graft infiltrating leukocytes (GIL) were flow sorted (7AAD⁻ CD45⁺) for bulk RNA sequencing. Gene set enrichment analysis was performed between GILs isolated from the ATG/CTLA4-Ig + αIL6 group [*n* = 4] and the ATG/CTLA4-Ig group [*n* = 4]. The tested gene sets are representative for genes specifically upregulated in Tregs but not effector T cells (Treg vs. Teff) **c** and for interleukin-4 and-13 (IL4/13) signaling and interleukin-10 (IL-10) signaling **d**.

Our RNA sequencing data revealed that signature genes for IL-10 signaling were significantly enriched within graft infiltrating leukocytes of αIL6 treated recipients, compared to the control group. In addition, the frequency of IL10⁺ CD4 cells was increased in the graft. IL-10 is a key mediator of an anti-inflammatory response[79]. In APC, IL-10 signaling downregulates MHC-II expression and cytokine secretion and thereby restrains antigen-specific T-cell proliferation. In Tregs, IL-10 mediated induction of FOXP3 and STAT3 is essential to maintain their suppressive function under inflammatory conditions[80,81]. In a feed-forward loop, IL-10 receptor signaling thereby also increases IL-10 secretion by Tregs[81]. Formally, we cannot discriminate whether the increased IL-10 signaling was the cause for or a consequence of more Tregs being present within cardiac allografts. Nonetheless, these results demonstrated that blocking IL-6 and the associated increase in intragraft Tregs favorably shaped the allograft microenvironment in a clinically relevant setting combining CTLA4-Ig and ATG. Depleting Tregs before or after transplantation led to graft loss. These data demonstrate that Treg-mediated regulation is a necessary mechanism to prevent CBRR under ATG/CTLA4-Ig + αIL6.

The role of intragraft inflammation in Treg migration was recently emphasized by a clinical trial in which low-dose IL-2 therapy triggered rejection and even graft loss in stable liver transplant recipients on CNI-based immunosuppression[82]. Interestingly, although a more than 2-fold expansion of peripheral blood Tregs was observed upon low-dose IL-2 therapy, Tregs did not accumulate within allografts. Rather, an IFNγ-driven transcriptional signature was dominant in liver biopsies 4 weeks after treatment initiation. The authors speculate that inflammation within the graft, driven by tissue-resident macrophages (Kupffer cells)[83], might have prevented intragraft Treg homing. We found significantly increased gene expression of CCL28 in grafts of αIL6 treated recipients. Thereby, CCR10 expressing Tregs could be recruited to the graft. Although the CCL28-CCR10 axis is not specific for Tregs, it could contribute to the observed increase in intragraft Treg levels[54]. Furthermore, we found that IL-6 blockade inhibits ser-oconversion of IgM to IgG DSA. This might contribute to the observed reduction in endothelial inflammatory receptor expression, as DSA specific for donor MHC on endothelial cells lead to endothelial acti-vation and enhanced trans-migration of effector T cells[84].

Targeting IL-6 or its receptor (IL-6R) has demonstrated ther-apeutic efficacy in several autoimmune and inflammatory diseases[85]. In clinical kidney transplantation, IL-6 signaling inhibition has been investigated mainly as a treatment of antibody-mediated allograft rejection (AMR)[86–90] and in de-sensitization protocols[91,92]. Notably, as part of a de-sensitization protocol, IL-6 blockade (clazakizumab) together with intravenous immunoglobulin and plasmapheresis resulted in a significant reduction in DSA and allowed kidney trans-plantation in 20/20 patients. Post-transplant, recipients received alemtuzumab induction and were maintained on tacrolimus-based

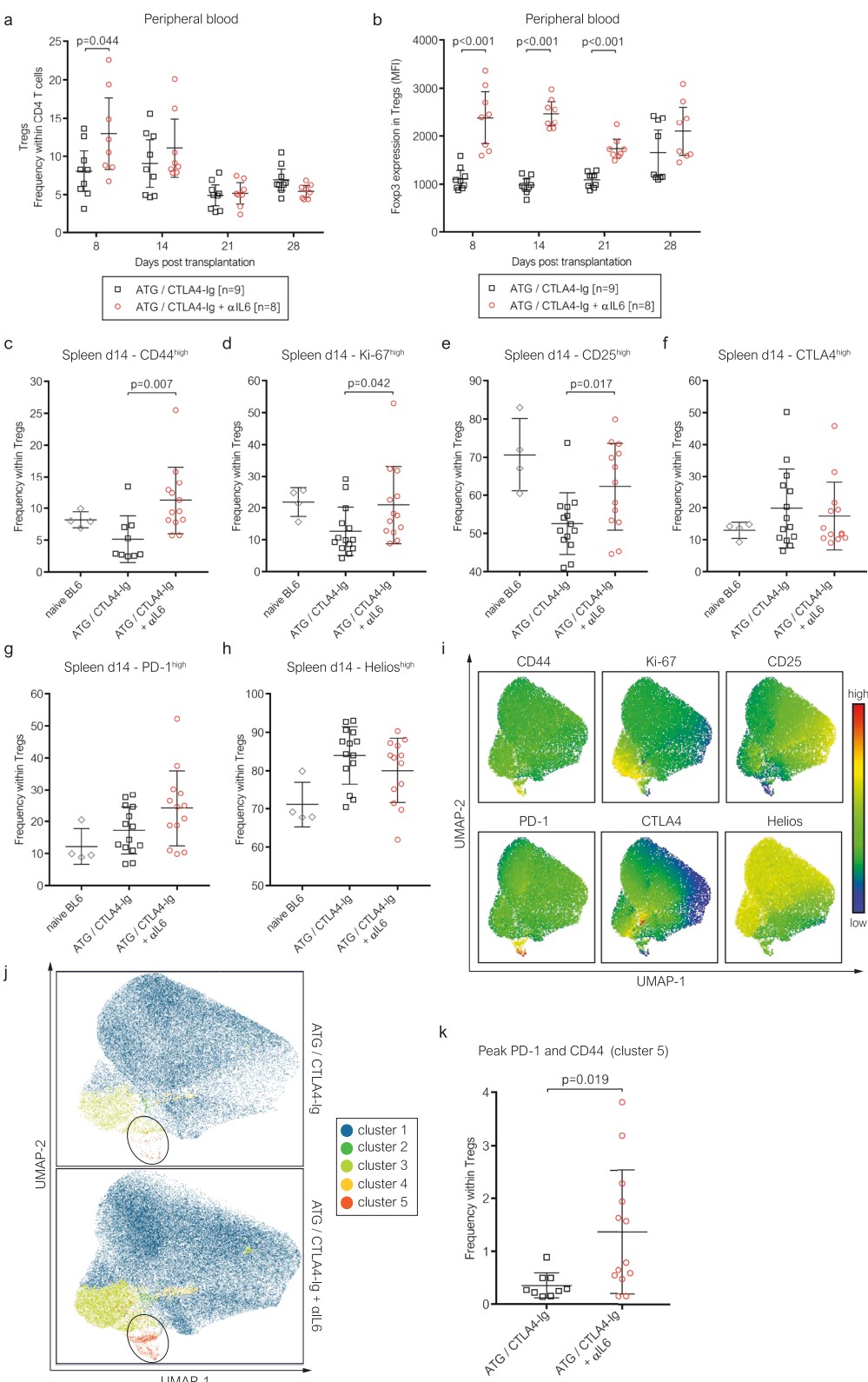

triple therapy plus monthly clazakizumab for the first 6-12 months. Interestingly, Treg frequencies 6- and 12-months post-transplant were significantly increased compared to baseline under this regimen[92]. While these encouraging reports indicate a potential benefit of IL-6/IL-6R blockade in AMR and de-sensitization, they rely on small sample sizes. A large phase III RCT investigating IL-6 blockade (clazakizumab) in chronic active AMR is currently ongoing (NCT03744910). Clinical data on IL-6 inhibition in de novo transplantation in non-sensitized recipients (i.e., as part of induction therapy) are not yet available with two trials investigating tocilizumab induction in cardiac (NCT03644667) and kidney transplantation (NCT04066114) currently ongoing.

The application of costimulation blockade in transplantation remains restricted by CBRR. Standard-of-care ATG induction protocols, which are successful in CNI-based regimens, fail to prevent CBRR[20,21]. Our results reveal insufficient intragraft regulation

**Fig. 5 | IL-6 blockade together with ATG specifically induces PD1^high CD44^high Tregs in the periphery. a–b** Peripheral blood regulatory T cells (Tregs, CD4^+ FOXP3^+) were quantified in cardiac allograft recipients via flow cytometry **a** (ATG/CTLA4-Ig [*n* = 9], ATG/CTLA4-Ig + αIL6 [*n* = 8]). Panel **b** depicts the mean FOXP3 expression (mean fluorescence intensity, MFI) in Tregs. **c–k** Spleens of cardiac allograft recipients were isolated 14 days after transplantation for phenotypical characterization of Tregs based on the expression of CD44 **c**, Ki-67 **d**, CD25 **e**, CTLA4 **f**, PD-1 **g** and Helios **h** (ATG/CTLA4-Ig [c: *n* = 9, **d**–**h**: *n* = 14], ATG/CTLA4-Ig + αIL6 [n = 13]). **i** UMAP projections generated using a concatenated file of 90,000 total Tregs (CD4^+ Foxp3^+) generated from 9 individual mice of the ATG/CTLA4-Ig group and 13 individual mice of the ATG/CTLA4-Ig + αIL6 group. The panels represent the density of Tregs expressing each phenotypical marker (CD44, Ki-67,

CD25, PD-1, CTLA4, Helios) as heat map statistic. **j** Automated clustering using FlowSOM identified 5 distinct Treg populations based on the expression profile of CD44, Ki-67, CD25, PD-1, CTLA4, and Helios. **k** The generated UMAP projections and clusters were applied to each individual Treg sample to compare the relative distribution of each cluster between the two treatment groups (ATG/CTLA4-Ig [*n* = 9], ATG/CTLA4-Ig + αIL6 [*n* = 13]). The circle in panel J indicates cluster 5 (red) within UMAP projections for each treatment group. Each symbol represents an individual mouse. All group comparisons (indicated with brackets) were conducted with a two-sided unpaired t-test. A correction for multiple testing was performed in panels **a** and **b**. Lines indicate group means ± SD. Representative flow cytometry data are provided in (suppl. Fig. 6).

associated with pro-inflammatory effects of ATG and identify IL-6 blockade as an effective and clinically available therapy that prevents ATG-resistant CBRR.

## Methods

### Laboratory animals
All experiments were approved by the local review board of the Medical University of Vienna and the Austrian Federal Ministry of Science, Research and Economy (vote number: BMWFW-66.009/0118-WF/V/3b/2016). Female C57BL/6 (H-2^b, strain code: 027) and BALB/c (H-2^d, strain code: 028) mice between 12 and 16 weeks of age were purchased from Charles River Laboratories (Germany) and Janvier-Labs (France). All mice were co-housed under barrier conditions in individually ventilated cages (up to 5 animals per cage) at 21 °C room temperature with a 12-hour light-dark cycle at the Core Facility Laboratory Animal Breeding and Husbandry at the Medical University of Vienna. Mice were handled in accordance with national and international guidelines of laboratory animal care and euthanized via cervical dislocation.

### Cardiac transplantation
Cervical heterotopic cardiac transplantation was performed as previously described[93]. In brief, the recipient's external jugular vein and common carotid artery were everted over a cuff under the microscope. Upon injection of 200IU heparin, the donor heart was harvested and flushed with 4 mL HTK solution (Custodiol, Koehler Chemie) through the aortic arch. The donor heart's pulmonary trunk and ascending aorta were anastomosed over the prepared cuff with the recipient's external jugular vein and common carotid artery, respectively. Graft survival was assessed by visual inspection and palpation for 100 days or until rejection. Rejection was defined as complete cessation of the heartbeat and was confirmed in histological analysis.

### Treatment regimen
CTLA-Ig (Abatacept, Bristol-Myers Squibb) was administered intraperitoneally according to the clinically approved dosing regimen (10 mg/kg body weight) corresponding to 0.25 mg/dose on days 0, 4, 14, 28, 56 and 84 after cardiac transplantation (day 0) (note: CTLA4-Ig/abatacept is used since belatacept is not binding to murine CD80 and CD86)[24]. A murine ATG (rabbit anti-mouse Ig) was provided by Sanofi-Genzyme. In brief, upon immunization of rabbits with murine thymocytes, serum was isolated, and the IgG fraction was isolated through column purification. ATG (6 mg/kg body weight, corresponding to 0.15 mg/dose) was administered i.p. on day 0 (immediately upon cross-clamp removal) and day 5. 0.6 mg of a blocking anti-IL6 mAb (clone MP5-20F3, BioXcell) were injected i.v. on day −1 and 0.3 mg on days 4 and 6. (Fig. 2a). Indicated groups of cardiac allograft recipients treated with ATG/CTLA4-Ig + αIL6 were injected with a depleting anti-CD25 mAB (clone: PC61; BioXcell, 0.25 mg/dose)[94] on days −5 and −2 (early Treg depletion) or on days 28 and 35 (late Treg depletion) in relation to cardiac transplantation (Fig. 4a). Treg depletion was confirmed via flow

cytometry using a second fluorophore-conjugated non-cross reacting anti-CD25mAB (clone: 7D4) (suppl. Fig. 3d).

### Histological analysis
Mice were sacrificed, and grafts were harvested at the time of rejection, at the end of follow-up (day 100), or 14 days after transplantation as indicated. Explanted cardiac allografts were fixed in 7.5% formaldehyde overnight and paraffin embedded. Sections were hematoxylin and eosin (HE) stained and scanned using an Aperio ScanScope scanner (Aperio Technologies). Grading was performed according to the International Society for Heart and Lung Transplantation (ISHLT) 2004 guidelines for cellular rejection score by an experienced transplantation pathologist (H.R.) blinded to the experimental background of the samples.

### Immunofluorescence microscopy
For CD8 and FOXP3 staining, FFPE sections of cardiac allografts were deparaffinized and rehydrated. Antigen retrieval was performed with Tris-EDTA buffer at pH9, and the samples were blocked with 5% BSA and 0.1% Tween20 in PBS. Tissue sections were stained with the primary antibodies (rat anti-mouse CD8, clone: 53-6.7, Biolegend and rabbit anti-mouse FOXP3, clone: EPR22102-37, Abcam) on 4 °C overnight and the secondary antibodies (AF488 polyclonal donkey anti-rat, Biolegend and AF555 goat anti-rabbit IgG H&L, Abcam) for 60 min at room temperature. For intracellular FOXP3 staining, 0.2% TritonX in PBS was used for permeabilization. DAPI (Biolegend) was used for nuclear staining at room temperature for 60 min. Image acquisition was performed using a Nikon Eclipse Ti microscope (Nikon). 9-16 pictures per area were acquired at a magnification of x60, corresponding to a mean area of 2.36 mm² analyzed per sample. The density of CD8 T cells and Tregs per mm² was quantified using NIS elements AR software (Nikon). The ratio between infiltrating Tregs and CD8 T cells was calculated.

For combined CD4, IL-10, and IL-17 staining, FFPE sections of cardiac allografts were deparaffinized and rehydrated. Antigen retrieval was performed with Tris-EDTA buffer at pH9. 0.2% TritonX in PBS was used for permeabilization. The samples were blocked with 5% nonfat dry milk and 0.1% Tween20 in PBS. Tissue sections were stained with a polyclonal rabbit anti-mouse IL17 antibody (Abcam), a rat anti-mouse IL-10 mAb (clone: JES5-2A5; Abcam), and a polyclonal goat anti-mouse CD4 antibody (R&D) on 4 °C overnight as primary antibodies. Donkey anti-rabbit (alexa-fluor 647), donkey anti-rat (alexa-fluor 488), and donkey anti-goat (alexa-fluor 555) IgG H&L were used as secondary antibodies and stained for 1 h on room temperature. DAPI (Biolegend) was used for nuclear staining at room temperature for 60 min. Image acquisition was performed using a tissue FAXs (TissueGnostics) slide scanner. One slide was scanned per sample, covering a mean area of 31.32 mm² (adjacent tissue or other artefact areas were excluded from analysis). CD4^+, IL-10^+, and IL17^+ cells were quantified using the high-plex fl module of HALO (indica labs). The number of CD4^+ IL-10^+ or CD4^+ IL-17^+ cells per mm² and the frequency of IL-10^+ or IL-17^+ cells within CD4^+ cells were calculated.

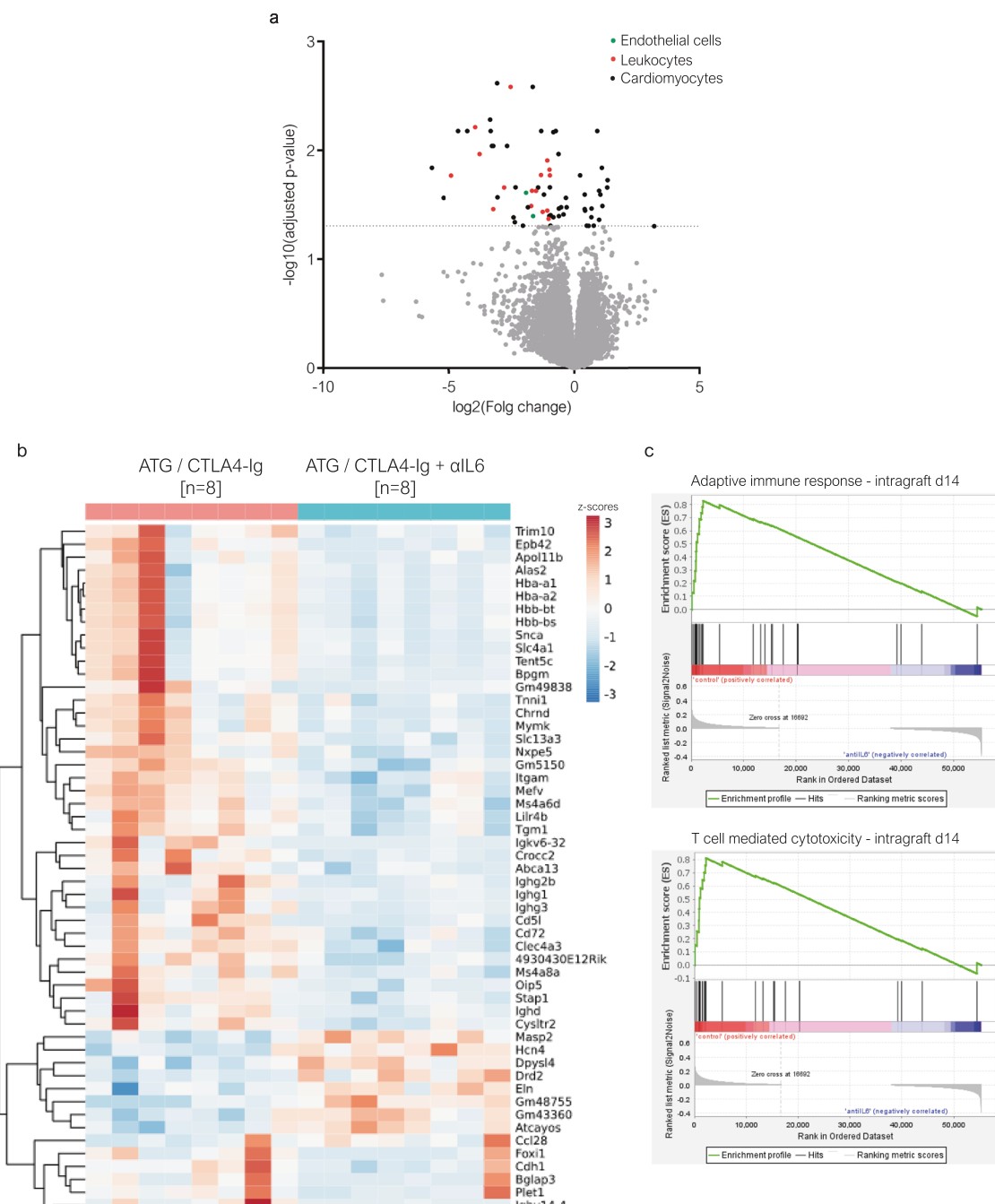

**Fig. 6 | IL-6 blockade attenuates the inflammatory response within cardiac allografts. a–c** C57BL/6 mice were grafted with a fully mismatched BALB/c heart under the indicated immunosuppressive regimen. Cardiac allografts were explanted 14 days after transplantation for histological and transcriptional analysis. RNA sequencing of paraffin-embedded cardiac allografts was performed. **a** Genes significantly up-or downregulated (within a Benjamini-Hochberg corrected Wald test) in the ATG/CTLA4-lg + αIL6 group ($n = 8$) compared to the ATG/CTLA4-lg ($n = 8$)

group are depicted as a volcano plot. Color codes indicate the expressing cell type. Panel **b** represents a heat map depicting all (protein-coding and non-coding) differentially expressed genes for each individual sample. Each individual cardiac allograft represents a column. **c** Gene set enrichment analysis was performed between ATG/CTLA4-Ig ("control", [$n = 8$]) and ATG/CTLA4-Ig + αIL6 ("antiIL6", [$n = 8$]) for gene sets representing an *adaptive* immune response (top) and T cell-mediated cytotoxicity (bottom).

## Flow cytometry

Single-cell suspensions of peripheral blood and spleen samples were prepared and stained for 30 min at 4 °C with flow cytometry antibodies (see Supplementary Table 1) upon red blood cell lysis. Intracellular staining was performed using a FOXP3/transcription factor staining buffer set (eBioscience). For analysis of graft infiltrating leukocytes, explanted cardiac allografts were enzymatically digested using a mouse tumor dissociation kit (Miltenyi) following

the manufacturer's instructions. Briefly, grafts were cut into small pieces and incubated for 40 min on 37 °C with an enzyme mixture. Residual red blood cells were lysed, and the samples were stained as described above. Absolute cell counts for GIL populations were calculated based on the total number of infiltrating leukocytes when whole hearts were digested. Flow cytometry was performed on a BD FACS Canto II and a BD LSR Fortessa. Data were analyzed using FlowJo (version 10.7.1; BD).

**Table 1 | Differentially expressed genes in cardiac allografts**

| Associated cell type/condition | Gene | Protein | Log2(Fold change) |
|---|---|---|---|
| Inflammatory response in cardiomyocytes and cardiac fibroblasts | *Eln* | Elastin | 1.31 |
| | *Mefv* | Mediterranean fever (Mefv), transcript variant 3 | −1.44 |
| | *Cysltr2* | Cysteinyl leukotriene receptor 2 | −1.07 |
| Cardiac distress/heart failure | *Foxi1* | Forkhead box I1 | −5.21 |
| | *Chrnd* | Cholinergic receptor, nicotinic, delta polypeptide | −4.62 |
| | *Hbb-bt* | Hemoglobin, beta adult t chain | −3.35 |
| | *Hbb-bs* | Hemoglobin, beta adult s chain | −3.32 |
| | *Hba-a2* | Hemoglobin alpha, adult chain 2 | −3.28 |
| | *Hba-a1* | Hemoglobin alpha, adult chain 1 | −3.22 |
| | *Alas2* | Aminolevulinic acid synthase 2 | −3.05 |
| | *Bpgm* | 2,3-bisphosphoglycerate mutase (Bpgm) | −2.34 |
| | *Cysltr2* | Cysteinyl leukotriene receptor 2 | −1.91 |
| | *Tgm1* | Transglutaminase 1 | −1.21 |
| Endothelial cells | *Ccl28* | CCL28 | 3.20 |
| | *Cysltr2* | Cysteinyl leukotriene receptor 2 | −1.91 |
| (graft infiltrating) Leukocytes | *Ighg1* | Immunoglobulinm heavy constant gamma 1 | −5.72 |
| | *Ighg3* | Immunoglobulinm heavy constant gamma 3 | −5.32 |
| | *EPB42* | Erythrocyte membrane protein band 4.2 | −4.90 |
| | *Ighg2b* | Immunoglobulinm heavy constant gamma 2b | −4.60 |
| | *Apol11b* | Apolipoprotein L 11b | −3.94 |
| | *Slc4a1* | Solute carrier family 4 | −3.76 |
| | *Trim10* | Tripartite motif-containing 10 | −3.23 |
| | *Snca* | Alpha synuclein | −2.79 |
| | *CD5L* | CD5 antigen like | −2.53 |
| | *Ighd* | Immunoglobulinm heavy constant delta | −1.93 |
| | *Oip5* | Opa Interacting Protein 5 | −1.84 |
| | *Ms4a8a* | CD20 | −1.71 |
| | *GM5150* | Predicted gene 5150 | −1.68 |
| | *Nxpe5* | Neurexophilin and PC-esterase domain family, member 5 | −1.66 |
| | *CD72* | CD72 | −1.52 |
| | *Ms4a6d* | Membrane-spanning 4-domains, subfamily A, member 6D (CD20 like protein) | −1.33 |
| | *Itgam* | Integrin alpha M (CD11b) | −1.25 |
| | *Stap1* | Signal transducing adaptor family member 1 | −1.08 |
| | *Clec4a3* | C-type lectin domain family 4 | −1.07 |
| | *Lilr4b* | Immunoglobulin-like receptor, subfamily B, member 4B | −1.02 |

RNA sequencing of paraffin-embedded cardiac allograft tissue isolated 14 days after transplantation was performed. Differentially expressed genes (DEG; log2 transformed fold change >1 and adjusted $p < 0.05$) between the ATG/CTLA4-Ig + αIL6 group ($n = 8$) and the ATG/CTLA4-Ig ($n = 8$) group are listed based on the expressing cell type and condition and ranked based on the log2 fold-change (ATG/CTLA4-Ig + αIL6 group vs. ATG/CTLA4-Ig).

**Donor-specific antibodies**

The serum of recipient mice was obtained at rejection or at the end of follow-up (day 100). After heat inactivation (30 min on 56 °C), 12.5 µl serum were incubated with $0.5 \times 10^6$ donor (BALB/c) thymocytes for 30 min on 37 °C. The binding of IgG on thymocytes was measured by staining with a fluorophore-conjugated anti-mouse pan-IgG antibody and reported as a percentage of IgG$^+$ thymocytes and median fluorescence intensity (MFI). Recipients were classified as "DSA positive" if the percentage of IgG$^+$ thymocytes exceeded a threshold defined as the mean of all negative controls (naïve BL6 serum) + 2 standard deviations.

**MHC-specific ELISA**

To measure donor-specific IgM and IgG antibodies against specific class I and II MHC antigens in the sera of transplanted mice, an ELISA was performed as described previously[95]. In brief, Nunc MaxiSorp ELISA plates (Thermo Fisher) were coated with recombinant MHC monomers, kindly provided by the NIH tetramer core facility (rH-2Dd, rI-Ad) at 5 µg/mL overnight. Sera were diluted 1:100 and incubated overnight. Bound antibodies were detected with rat anti-mouse IgG1 (clone A85-1), IgG2a (clone R19-15), IgG2b (clone R9-91), IgG3 (clone R2-38) or IgM (clone R6-60.2) mAbs (BD) diluted 1:1000, and a 1:2000 diluted horseradish peroxidase (HRP)−coupled goat anti-rat antiserum (Amersham Biosciences). The substrate for HRP was ABTS. Absorption was measured with a Victor microplate reader (PerkinElmer) at 405-492 nm.

**Cytokine multiplex assay**

For serum cytokine measurements, the Legendplex mouse inflammation panel (IL-1α, IL-1β, IL-6, IL-10, IL-12p70, IL-17A, IL-23, IL-27, MCP-1, IFN-β, IFN-γ, TNF-α, and GM-CSF, Biolegend) was used following the manufacturer's instructions. In brief, serum samples were incubated with capture beads on 4 °C overnight on a plate shaker. Upon washing, biotinylated detection antibodies and PE-streptavidin were added.

Samples were analyzed on a BD LSR Fortessa. For each cytokine, an 8-point standard curve was constructed to convert the mean fluorescence intensities (MFI) to concentrations in pg per ml serum. All samples were analyzed in duplicates.

## RNA sequencing

For RNA sequencing of cardiac allograft samples, RNA was extracted from formaldehyde-fixed paraffin-embedded (FFPE) tissue blocks using an RNeasy FFPE kit (Qiagen) according to the manufacturer's instructions. Tissue sample integrity was confirmed via HE histology. Thirty 5 μm sections per sample were used for RNA extraction. RNA integrity was assessed on a Bioanalyzer 2100 (Agilent) using an RNA 6000 Nano kit. 16 sequencing libraries were prepared at the Core Facility Genomics, Medical University of Vienna, using the NebNext Ultra II Directional RNA Library Prep Kit (E7760) and NEBNext rRNA Depletion Kit v2 (E7400) according to manufacturer's protocols (New England Biolabs).

For RNA sequencing of isolated graft infiltrating leukocytes (GILs), freshly explanted cardiac allografts were enzymatically digested as described above. Graft infiltrating leukocytes were flow sorted as 7AAD⁻ CD45⁺ cells on a BD FACSAria Fusion (BD). RNA was isolated from purified GILs using an RNeasy kit (Qiagen) following the manufacturer's instructions. RNA integrity was assessed on a Bioanalyzer 2100 (Agilent) using an RNA 6000 Nano kit. 8 sequencing libraries were prepared at the Core Facility Genomics, Medical University of Vienna, using the NEBNext Single Cell/Low Input RNA Library Prep Kit (E6420) according to manufacturer's protocols (New England Biolabs).

Libraries were QC-checked on a Bioanalyzer 2100 (Agilent) using a high sensitivity DNA kit for correct insert size and quantified using Qubit dsDNA HS Assay (Invitrogen). Pooled libraries were sequenced on two flow cells of a NextSeq500 instrument (Illumina) in 1x75bp single-end sequencing mode. On average 38.5 million reads were generated per sample. Reads in fastq format were aligned to the mouse reference genome version GRCm38[96] with Gencode mV23 annotations[97] using STAR aligner[98] version 2.6.1a in 2-pass mode. Reads per gene were counted by STAR, and differential gene expression was calculated using DESeq2[99] version 1.22.2. TPM were generated by RSEM[100].

## Statistical analysis

Ordinal variables were compared between two groups using a two-sided Mann-Whitney U test (ISHLT scores) and Fisher's exact test (DSA positive/negative recipients). Normal distributed continuous variables were compared between two groups with two-sided unpaired t-tests and are depicted as individual values with lines and error bars indicating group means ± SD. Continuous variables that did not follow a normal distribution (graft infiltrating CD4⁺ IL10⁺ and CD4⁺ IL17⁺ cells per mm² and serum cytokine levels) (Fig. 1f, suppl. Fig. 2b, Suppl. Fig. 4b, d) were compared between two groups using a two-sided Mann-Whitney-U test and are depicted as individual values with lines indicating the group median ± IQR. Correction for multiple testing was performed in case one hypothesis was tested at multiple time points in the same individuals (Fig. 5a, b) using the Bonferroni-Holm method. Cardiac allograft survival was compared between two groups using a two-sided log-rank test. Differential gene expression between two groups was assessed using a Wald test and the Benjamini-Hochberg method to correct for multiple testing. Gene set enrichment analysis (GSEA) was performed as previously described using the open-source GSEA software[101,102] and curated gene sets from the molecular signature database (MSigDB codes: MM4209, MM4278, M27382). A two-tailed $p$ value below 0.05 was considered statistically significant.

## Reporting summary

Further information on research design is available in the Nature Portfolio Reporting Summary linked to this article.

## Data availability

The datasets generated during and/or analyzed during the current study are available from the corresponding author on request. RNA sequencing data that support the findings of this study are deposited in the NCBI GEO/SRA repository and are publicly available under the accession number GSE241472. Source data are provided with this paper.

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

## Acknowledgements

Recombinant MHC monomers were kindly provided by the NIH Tetramer Core Facility. The graphical illustrations in panels 1a, 2a, and 4a were designed using Biorender (publication licenses: YL259RBWA4, RS259RBQHM, and UE26GKIC5E). The RNA sequencing data presented within this manuscript were obtained in collaboration with the Core Facility Genomics of the Medical University of Vienna, a member of Vienna Life-Science Instruments. The imaging data presented in supplementary Fig. 4 were acquired in collaboration with the Core Facility Imaging of the Medical University of Vienna. This work was funded by the

Medical Scientific Fund of the Mayor of Vienna through Project number 21050 (to MM) and project number 22225 (to TW). Additional funding from Vienna Science and Technology Fund (WWTF) through project LS18-031 (to TW) and from the Country of Lower Austria Danube Allergy Research Cluster (DARC) grant (to TW). AMW is a recipient of a DOC Fellowship of the Austrian Academy of Sciences (DOC/25556).

## Author contributions

T.W. and M.M. designed the experiments, analyzed the data and wrote the manuscript. K.M., A.W., R.S., V.K., M.B., and A.K. assisted in experiments. H.R. provided the histological analysis of cardiac tissue sections. S.D. processed RNA sequencing data and provided bioinformatics support.

## Competing interests

TW received speaker's honoraria from eGenesis and Mallinckrodt/Therakos and is a DSMB member for Quell Therapeutics. The other authors have no competing interests to disclose.
