## [Peer Review File · Nature Communications]

IL-6 inhibition prevents costimulation blockade-resistant allograft rejection in T cell-depleted recipients by promoting intragraft immune regulation in miceReviewer #1 (expert in cardiac transplantation):

The authors demonstrate that adding anti-IL-6 therapy to ATG and CTLA4-Ig treatments improves murine heart allograft survival to 100 days post-transplantation. This is associated with a reduction in intra-graft CD8+ TEM cells and an increase in intra-graft Tregs, when compared to ATG/CTLA4-Ig treatment alone. The results are clear and well presented.

My major concern is that instead of specifically targeting the mechanisms underlying costimulation blockade-resistant rejection, the addition of IL-6 signaling may have simply reached a threshold of general immunosuppression that achieved 100 day survival. To better support their conclusion, the authors should consider 1) adding a "third-party" immunosuppressant to ATG and CTLA4-Ig which is Treg-unfriendly (CyA), and 2) providing survival data in recipients treated with ATG and anti-IL-6 alone. In addition, to better support the conclusion that intra-graft Tregs specifically are preventing costimulation-resistant rejection, the authors should consider repeating the triple drug studies and with Treg depletion.

Reviewer #2 (expert in T cells in transplantation and immunosuppression):

The manuscript by Wekerle and colleagues explores the concept of blocking inflammatory signals (IL-6) to promote long-term graft survival in the setting of costimulatory blockade. The strengths of the manuscript are the rigor of the data presented and clarity of the hypotheses and rationale, as well as the novel conceptual approach of blocking innate immune signals as an adjunct immunotherapy along with costimulation blockade. Moreover, the use of automated clustering analysis to identify subsets of Treg expanded following IL-6 blockade (PD-1hi CD44hi) is novel and advances the field forward.

The main limitation of the study is the limited mechanistic depth as far as what effects of IL-6 blockade are responsible for the observed improvement in allograft survival. The blockade is associated with an increase in Treg in the graft but no experiments are done to test the requirement for Treg in the observed prolongation in graft survival.

A few other points would increase the clarity of the manuscript:

1. The elicitation of inflammatory cytokines as a result of T cell depletion using ATG is presented as the rationale for using IL-6 blockade. Surgical trauma can of course also drive inflammatory cytokine responses. However IL-6 blockade had no effect in recipients treated with CTLA-4 Ig alone, without the addition of ATG. Can the authors discuss why this is the case?
2. The serum cytokine analysis following ATG shows that the most strongly and consistently elicited inflammatory cytokine was IFN- β . What was the rationale for pursuing anti-IL-6 as a strategy instead of anti-IL-1 β ?
3. IL-6 is well known to drive IL-17 responses. The authors reported increased Treg and IL-4/IL-10 (Th2) differentiation; could they comment on the impact of IL-6 blockade on Th17 differentiation under these conditions?

Reviewer #3 (expert in transplantation immunology and IL-6 blockade):

This very interesting paper by Wekerle et al. details a carefully analyzed approach to understanding costimulatory blockade resistant rejection (CSBRR) in a well established mouse model of cardiac allograft injury. This is significant since many of the issues noted in the mouse model are common to the human transplantation where early use of CTLA4-Ig post transplant is commonly associated with severe rejection episodes, likely mediated by CD8+Teffector/Memory (CD8TEM) cells that are CD28 negative and thus resistant to CTLA4-Ig blockade. Initial induction therapy was with anti-IL-2R which resulted in unacceptable high rates of rejection (50-60%) in some centers when used without CNIs. There were attempts to modify this protocol using T-cell depletion with ATG or Campath, but early rejection episodes were still seen. This is unfortunate, since CTLA4-Ig used in patients who are maintained on Tacrolimus for 6M-12M and gradually

tapered off can yield great benefits to the patients. However, no one has figured out how to use CTLA4-ig w/o CNIs early. The authors recapitulated these findings in their mouse model and noted that despite T-cell depletion, grafts were still lost from rejection and that the addition of ATG to CTLA-4Ig while reducing Teffector cells and increasing Tregs in peripheral blood did not mitigate inflammation and CD8T-effector cells in the allograft. The authors noted that IL-6 was released after ATG administration and then added anti-IL-6 treatment to the ATG+ CTLA4-Ig. This appeared to correct the CSBRR and improve long-term graft survival. Importantly, anti-IL-6 reduced the inflammation and CD8TEM cells in the allograft and peripheral blood and increased high affinity Tregs. A reduction in DSAs was also seen. Overall, these are critical experiments that could help change immunosuppression protocols for all transplant patients. It suggests we can possibly move from a current utility of immunosuppressive agents to a more tolerable protocol of immunomodulation. This paper is informative, but I think the discussion could be improved by discussing this possibility. Importantly, the authors relate that there are several ongoing studies of anti-IL-6 /IL-6R as induction agents in human transplantation, but none have results to date. This isn't completely true as the paper by Vo et al (ref 88) describing the use of clazakizumab (anti-IL-6) as a desensitization agent in highly-HLA sensitized patients showed that anti-IL-6 treatment resulted in significant reductions in HLA antibodies and allowed 20/20 patients to be transplanted. Importantly, transplant patients received T-cell depletion with Campath and were continued on anti-IL-6 monthly with decreased dose tacrolimus and cellcept. This resulted in reduction/elimination of pre-transplant DSAs and importantly significant increases in peripheral Tregs at 6-12M post-transplant. This should be mentioned as supportive human data.

I think the paper is clear and well written. The experimental design is acceptable and appropriate controls are included.

Point-by-point response

NCOMMS-23-38498

Response to reviewer 1:

- *“The authors demonstrate that adding anti-IL-6 therapy to ATG and CTLA4-Ig treatments improves murine heart allograft survival to 100 days post-transplantation. This is associated with a reduction in intra-graft CD8+ TEM cells and an increase in intra-graft Tregs, when compared to ATG/CTLA4-Ig treatment alone. The results are clear and well presented.*

My major concern is that instead of specifically targeting the mechanisms underlying costimulation blockade-resistant rejection, the addition of IL-6 signaling may have simply reached a threshold of general immunosuppression that achieved 100 day survival. To better support their conclusion, the authors should consider 1) adding a “third-party” immunosuppressant to ATG and CTLA4-Ig which is Treg-unfriendly (CyA)”

We thank the reviewer for the thorough assessment of our manuscript. We acknowledge that a general immunosuppressive effect of α IL-6 might theoretically contribute to the prolonged cardiac allograft survival in our model. We think that depleting Tregs (as suggested by the reviewer in a later comment) is the most specific approach to determine whether promoting Treg-mediated regulation or adding to general immunosuppression is the primary mechanism by which α IL6 prolongs graft survival in our model. Hence, we conducted additional experiments in which we depleted Tregs in cardiac allograft recipients treated with ATG/CTLA4-Ig + α IL6 (**new Figure 4a and 4b**). Treg depletion before transplantation led to graft loss within 30 days in the majority of recipients. Similarly, Treg depletion 4 weeks after transplantation, when ATG and α IL6 have reached subtherapeutic levels, also abrogated cardiac allograft survival. These data demonstrate that the effect of α IL6 in our model is Treg-dependent.

- *“, and 2) providing survival data in recipients treated with ATG and anti-IL-6 alone.”*

We agree that the requested data are relevant for the manuscript. Therefore, we now include the survival data of this important control group receiving a cardiac allograft under ATG + α IL6 (without maintenance immunosuppression) in **Figure 2b**. We also added survival data of recipients treated with α IL6 only. Under α IL6, grafts were rejected with a similar kinetic to untreated recipients. When combined with ATG (ATG + α IL6) graft survival was prolonged compared to either substance administered individually, but eventually almost all grafts were rejected by day 45.

- *“In addition, to better support the conclusion that intra-graft Tregs specifically are preventing costimulation-resistant rejection, the authors should consider repeating the triple drug studies and with Treg depletion.”*

We agree that the suggested experiments are highly valuable to support our main conclusion. As described above, we carried out experiments in which we depleted Tregs under ATG/CTLA4-Ig + α IL6. Treg depletion led to graft loss, supporting our conclusion that IL-6 blockade prevents CBRR via Treg-mediated regulation.

Response to reviewer 2:

- *“The manuscript by Wekerle and colleagues explores the concept of blocking inflammatory signals (IL-6) to promote long-term graft survival in the setting of costimulatory blockade. The strengths of the manuscript are the rigor of the data presented and clarity of the hypotheses and rationale, as well as the novel conceptual approach of blocking innate immune signals as an adjunct immunotherapy along with costimulation blockade. Moreover, the use of automated clustering analysis to identify subsets of Treg expanded following IL-6 blockade (PD-1hi CD44hi) is novel and advances the field forward.”*

“The main limitation of the study is the limited mechanistic depth as far as what effects of IL-6 blockade are responsible for the observed improvement in allograft survival. The blockade is associated with an increase in Treg in the graft but no experiments are done to test the requirement for Treg in the observed prolongation in graft survival.”

We thank the reviewer for the constructive comments on our manuscript. We acknowledge that additional data demonstrating the requirement for Tregs in the observed prolongation of graft survival are important. We therefore conducted experiments, in which we depleted Tregs before or 4 weeks after transplantation (in recipients treated with ATG/CTLA4-Ig + α IL6). At both time points, Treg depletion led to graft loss, supporting the conclusion that Tregs are required for α IL6-mediated prolongation of cardiac allograft survival (see also response to reviewer 1) (**new Figure 4a and 4b**).

- *“A few other points would increase the clarity of the manuscript: 1. The elicitation of inflammatory cytokines as a result of T cell depletion using ATG is presented as the rationale for using IL-6 blockade. Surgical trauma can of course also drive inflammatory cytokine responses. However, IL-6 blockade had no effect in recipients treated with CTLA-4 Ig alone, without the addition of ATG. Can the authors discuss why this is the case?”*

This is an interesting point. Indeed, blocking IL-6 improved graft survival only when combined with ATG, but not when combined with CTLA4Ig (without ATG) or given alone (**new data in Figure 2**), although inflammatory cytokines are also secreted independent of ATG (triggered by ischemia reperfusion injury and surgical trauma). These data suggest that in our model neutralizing IL-6-driven inflammation in a non-depleted T cell compartment is not sufficient to prevent rejection under chronic CTLA4-Ig. Instead, we think that ATG and α IL6 act synergistically (**new data in Figure 2**): ATG creates a peripheral T cell compartment that favors Tregs. However, we show that under inflammatory conditions, Tregs, although enriched peripherally, fail to accumulate within the graft. Blocking IL-6 resolved this imbalance between peripheral and intragraft Treg levels and prevented CBRR. We therefore conclude that the critical contribution of α IL6 in our model is to extend Treg-mediated control over donor-reactive T cells from the periphery to the intragraft compartment. We included this point in our discussion accordingly (Page 13, lines 348-356).

- *“2. The serum cytokine analysis following ATG shows that the most strongly and consistently elicited inflammatory cytokine was IFN-beta. What was the rationale for pursuing anti-IL-6 as a strategy instead of anti-IL-1beta?”*

Recent data demonstrated that IFN-beta stabilizes Tregs via foxp-3 acetylation (PMID: 35148827). Interestingly, in this report, exogenous IFN-beta administration synergized with CTLA4-Ig in preventing cardiac allograft survival. Therefore, IFN-beta was not the preferred therapeutic target to be blocked for us. Our data did not show a consistent increase in IL-1-beta levels when ATG was administered in the presence of CTLA4-Ig (**Figure 1e**). We found 4 cytokines to be induced by ATG: IL-23, IFN-beta, IFN-gamma, and IL-6. We decided to pursue IL-6 as therapeutic target due to its ability to deliver inflammatory signals broadly via trans-signaling (using a soluble IL-6 receptor and ubiquitously expressed gp130) and its well-established negative role in peripheral Treg induction. The availability of clinically approved antibodies targeting IL-6 also made it an attractive target from the perspective of potential translatability. Furthermore, since IL-6 is typically induced early in the course of an inflammatory response, we suspected that blocking IL-6 might indirectly also dampen the release of other downstream inflammatory mediators. We now included new data revealing that indeed neutralizing IL-6 led to decreased serum levels of most inflammatory mediators (including the ones induced by ATG) 7 days after transplantation (**new suppl. Figure 2a and 2b**).

- *“3. IL-6 is well known to drive IL-17 responses. The authors reported increased Treg and IL-4/IL-10 (Th2) differentiation; could they comment on the impact of IL-6 blockade on Th17 differentiation under these conditions?”*

We agree that the potential impact of blocking IL-6 on Th17 responses in our model is a relevant issue, especially since Th17 cells have been implicated in CBRR (PMIDs: 24493820, 24730049).

We therefore quantified IL-10 and IL-17 secreting cells within cardiac allografts explanted 14 days after transplantation via immunofluorescent microscopy (**new suppl. Figure 4**). Blocking IL-6 increased the frequency (within CD4+ cells) and absolute number (cells/mm²) of CD4+ IL10+ cells within cardiac allografts. However, IL-6 blockade had no effect on the frequency (within CD4+ cells) or absolute number (cells/mm²) of CD4+ IL-17+ cells within cardiac allografts. Furthermore, genes associated with IL-17 signaling were not enriched in either treatment group in a gene set enrichment analysis (nominal p=0.334). These data indicate that impairing Th17 responses is not a critical mechanism by which IL-6 blockade prolongs cardiac allograft survival in our model. We included these findings in our manuscript (page 10, lines 227-240).

Response to reviewer 3:

- *“This very interesting paper by Wekerle et al. details a carefully analyzed approach to understanding costimulatory blockade resistant rejection (CSBRR) in a well established mouse model of cardiac allograft injury. This is significant since many of the issues noted in the mouse model are common to the human transplantation where early use of CTLA4-Ig post transplant is commonly associated with severe rejection episodes, likely mediated by CD8+Teffector/Memory (CD8TEM) cells that are CD28 negative and thus resistant to CTLA4-Ig blockade. Initial induction therapy was with anti-IL-2R which resulted in unacceptable high rates of rejection (50-60%) in some centers when used without CNIs. There were attempts to modify this protocol using T-cell depletion with ATG or Campath, but early rejection episodes were still seen. This is unfortunate, since CTLA4-Ig used in patients who are maintained on Tacrolimus for 6M-12M and gradually tapered off can yield great benefits to the patients. However, no one has figured out how to use CTLA4-Ig w/o CNIs early. The authors recapitulated these findings in their mouse model and noted that despite T-cell depletion, grafts were still lost from rejection and that the addition of ATG to CTLA-4Ig while reducing Teffector cells and increasing Tregs in peripheral blood did not mitigate inflammation and CD8T-effector cells in the allograft. The authors noted that IL-6 was released after ATG administration and then added anti-IL-6 treatment to the ATG+ CTLA4-Ig. This appeared to correct the CSBRR and improve long-Term graft survival. Importantly, anti-IL-6 reduced the inflammation and CD8TEM cells in the allograft and peripheral blood and increased high affinity Tregs. A reduction in DSAs was also seen. Overall, these are critical experiments that could help change immunosuppression protocols for all transplant patients. It suggest we can possibly move from a current utility of immunosuppressive agents to a more tolerable protocol of immunomodulation. This paper is informative, but I think the discussion could be improved by discussing this possibility.”*

We thank the reviewer for the helpful comments and agree that a summarizing conclusion would strengthen the manuscript. We modified our discussion accordingly (page 13/14, lines 356-360).

- *“Importantly, the authors relate that there are several ongoing studies of anti-IL-6 /IL-6R as induction agents in human transplantation, but none have results to date. This isn't completely true as the paper by Vo et al (ref 88) describing the use of clazakizumab (anti-IL-6) as a desensitization agent in highly-HLA sensitized patients showed that anti-IL-6 treatment resulted in significant reductions in HLA antibodies and allowed 20/20 patients to be transplanted. Importantly, at transplant patients received T-cell depletion with Campath and were continued on anti-IL-6 monthly with decreased dose tacrolimus and cellcept. This resulted in reduction/elimination of pre-transplant DSAs and importantly significant increases in peripheral Tregs at 6-12M post-transplant. This should be mentioned as supportive human data.”*

This is a valuable suggestion. We adapted our discussion to include the abovementioned clinical data (page 14/15, lines 392-398).

- *“I think the paper is clear and well written. The experimental design is acceptable and appropriate controls are included.”*

We thank the reviewer for the positive comments on our work.

General comments:

- While revising the submitted manuscript, we found that the serum cytokine levels presented in Figure 1f were by mistake analyzed parametrically despite not following a normal distribution. We regret the mistake and corrected the analysis accordingly. The resulting change in p values did not influence the conclusion drawn from the presented data (as the results remained significant at a $p=0.0079$ after the correction). We indicated the use of a non-parametric Mann-Whitney U test in the figure legend and the methods section.
- Marlena Buresch contributed to the experiments performed for the revision and was therefore added as co-author to the manuscript.
- To integrate the new data generated for this revision in the revised manuscript, the original figure 3 was re-arranged, a new figure 4 was included and the subsequent figures were re-numbered accordingly. Supplementary figures 2 and 3 were re-arranged and a new supplementary figure 4 was included. References and discussion were updated to be aligned with the revisions made.

Reviewer #1 (Remarks to the Author):

The authors did an excellent job responding the reviewer's comments and generated a much-improved manuscript.

Reviewer #2 (Remarks to the Author):

The authors have thoroughly responded to the critiques raised during the initial review, in most cases with the addition of new experimental data to better support the conclusions reached in the manuscript. In particular, the inclusion of new data showing the results of Treg depletion better supports the conclusion that Tregs are involved in the mechanism of anti-IL-6 mediated prolongation of graft survival. The findings that ATG is required for this effect suggests that Treg are generated during homeostatic reconstitution in the setting of anti-IL-6, which is a novel finding that moves the field forward. Finally, the addition of new data ruling out a role for decreased Th17 differentiation in the context of anti-IL-6 is informative. Overall, the revised manuscript is rigorous, comprehensive, and contributes important insight regarding the mechanisms underlying costimulation blockade-resistant rejection during transplantation.

Reviewer #3 (Remarks to the Author):

This paper by Wekerle represents a revised manuscript regarding the role of IL-6 blockade in co-stimulatory blockade resistant rejection (CSBRR). The authors have done an excellent job in revising and improving the original manuscript. From my standpoint, this is a very important paper with significant implications for clinical care of transplant recipients as is outlined in the manuscript. I think the revisions have resulted in a more focused and clear presentation of the data and flows in a logical and focused way. The authors demonstrate that IL-6 is a cytokine critical for CSBRR and blockade of IL-6 + CTLA4Ig and ATG results in excellent survival with induction of Tregs. The maintenance of this rejection free state is dependent on Tregs induced by ATG + anti-IL-6. From a critique standpoint, I feel the authors could improve the figures as they are in pale print and hard to read. Also the authors should update the statement in the discussion regarding the IMAGINE study as it has now been terminated due to lack of efficacy. I do not think this has relevance to the data the authors are presenting as many feel this study failed due to under dosing of anti-IL6. Overall, excellent paper that should lead to larger studies in human organ transplantation.

Point-by-point response

NCOMMS-23-38498A

Response to reviewer 1:

- *“The authors did an excellent job responding the reviewer's comments and generated a much-improved manuscript.”*

We thank the reviewer for the positive assessment of our revised manuscript.

Response to reviewer 2:

- *“The authors have thoroughly responded to the critiques raised during the initial review, in most cases with the addition of new experimental data to better support the conclusions reached in the manuscript. In particular, the inclusion of new data showing the results of Treg depletion better supports the conclusion that Tregs are involved in the mechanism of anti-IL-6 mediated prolongation of graft survival. The findings that ATG is required for this effect suggests that Treg are generated during homeostatic reconstitution in the setting of anti-IL-6, which is a novel finding that moves the field forward. Finally, the addition of new data ruling out a role for decreased Th17 differentiation in the context of anti-IL-6 is informative. Overall, the revised manuscript is rigorous, comprehensive, and contributes important insight regarding the mechanisms underlying costimulation blockade-resistant rejection during transplantation.”*

We thank the reviewer for the positive comments on our work.

Response to reviewer 3:

- *“This paper by Wekerle represents a revised manuscript regarding the role of IL-6 blockade in co-stimulatory blockade resistant rejection (CSBRR). The authors have done an excellent job in revising and improving the original manuscript. From my standpoint, this is a very important paper with significant implications for clinical care of transplant recipients as is outlined in the manuscript. I think the revisions have resulted in a more focused and clear presentation of the data and flows in a logical and focused way. The authors demonstrate that IL-6 is a cytokine critical for CSBRR and blockade of IL-6 + CTLA4Ig and ATG results in excellent survival with induction of Tregs. The maintenance of this rejection free state is dependent on Tregs induced by ATG + anti-IL-6. From a critique standpoint, I feel the authors could improve the figures as they are in pale print and hard to read.*

We thank the reviewer for the constructive comments and provide revised figures with increased text size.

- *Also the authors should update the statement in the discussion regarding the IMAGINE study as it has now been terminated due to lack of efficacy. I do not think this has relevance to the data the authors are presenting as many feel this study failed due to under dosing of anti-IL6.*

We agree that a termination of the IMAGINE trial would need to be mentioned. Yet, on [clinicaltrials.gov](https://clinicaltrials.gov/study/NCT03744910) (<https://clinicaltrials.gov/study/NCT03744910>) the trial is still registered as active without any statement regarding its termination. Further, we could not find a publicly available source to confirm the termination of the trial due to lack of efficacy. Hence, as the information is not publicly available at the time of this writing, we cannot state that the trial was terminated within our manuscript.

- *Overall, excellent paper that should lead to larger studies in human organ transplantation.*

We thank the reviewer for the positive review of our manuscript.